# GENERATING GFLOWNETS AS YOU WISH WITH DIFFUSION PROCESS

## ABSTRACT

Generative Flow Networks (GFlowNets) are probabilistic samplers that learn stochastic policies to generate diverse sets of high-reward objects, which is essential in scientific discovery tasks. However, most existing GFlowNets necessitate training, becoming costly as the diversity of GFlowNets expands and trajectory lengths increase. To alleviate this problem, we propose a method to **Gen**erate high-performing **GFlowNet** parameters based on a given model structure, called **GenFlowNet**. Specifically, we first prepare an autoencoder to extract latent representations of GeFlowNet parameters and reconstruct them. Then, a structure encoder is trained alongside a conditional latent diffusion model to generate the target GFlowNet parameters based on the given structure information. To the best of our knowledge, it is the first exploration to generate parameters of a probabilistic sampler using the diffusion process. It enables us to obtain a new GFlowNet without training, effectively reducing the trial-and-error cost during GFlowNet development. Extensive experiments on diverse structures and tasks validate the superiority and generalizability of our method.

## 1 INTODUCTION

Generative Flow Networks (GFlowNets) rooted in foundational theoretical work (Bengio et al., 2023; Lahlou et al., 2023) and closely linked to variational inference (Malkin et al., 2022; Zimmermann et al., 2022; Zhang et al., 2024), represent a class of methods for sampling discrete objects from multimodal distributions. Owing to their flexibility, GFlowNets have been effectively applied to a wide range of problems where diverse high-quality candidates are needed, such as molecules (Bengio et al., 2021) and biological sequences (Jain et al., 2022).

They typically learn a generative policy to sample discrete objects $x$ with a non-negative reward $R(x)$. Most existing methods are trained using stochastic gradient descent (SGD) to optimize the learning objective on states or trajectories sampled from a training policy (Shen et al., 2023). Standard learning objectives ensure that the GFlowNet's learned distribution over $x$ matches the target distribution $p(x) \propto R(x)$ when the training loss is globally minimized across all states or trajectories. However, *GFlowNet can hardly ascertain the performance in advance, including those across different tasks, without training.* In many practical scenarios, the object space and the associated trajectories can be exponentially large, which significantly increases the training burden of GFlowNets.

A promising solution to overcome these burdens is to bypass the training phase of GFlowNets and directly obtain the high-

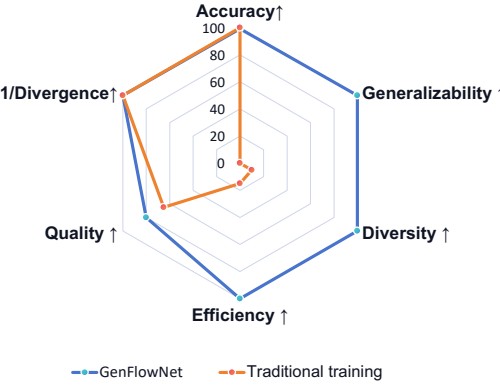

Figure 1: GenFlowNet demonstrates superior performance over the traditional training paradigm across multiple dimensions: accuracy, diversity, training cost efficiency, generalizability, and overall quality.

performing parameters tailored to downstream tasks. This has been explored in previous parameter generation techniques (Peebles et al., 2022; Erkoç et al., 2023; Schürholt et al., 2022; Jin et al., 2024), which can generate new parameters without training. However, these methods are specifically tailored for traditional neural networks, not probabilistic samplers like GFlowNets. *The distinction between traditional neural networks and GFlowNets makes it challenging to adapt these methods for generating GFlowNet parameters.* Moreover, they lack the flexibility to adapt to new conditions, *failing to generalize to unknown structures and tasks during training, which hinders their practical application.* Therefore, it is critical to develop a parameter generation method for GFlowNets that can generalize to diverse structures and tasks.

In this paper, we introduce a method to **Gen**erate **GFlowNet** parameters based on model structures, termed as **GenFlowNet**. It comprises an autoencoder (AE) and a conditional diffusion model designed to capture the distribution of GFlowNet parameters with specified structures. Initially, the AE is trained to compress and reconstruct GFlowNets parameters across various structures. Subsequently, we represent GFlowNets via their structure information, which is then converted into embeddings by a structure encoder. These embeddings serve as conditioning inputs for a conditional diffusion model, which generates latent parameter representations from a noise distribution. During inference, given a specific structure, the diffusion model, in conjunction with the AE decoder, can synthesize the desired GFlowNet parameters.

Extensive experiments on diverse structures and tasks demonstrate the following key characteristics of our method (see Fig. 1):

- Generalizability: Our method can generate GFlowNet parameters for various unknown structures and tasks during inference.

- High Performance: The parameters produced by our method achieve performance comparable to or exceeding that of GFlowNets trained with traditional paradigms.

- Diversity and Efficiency: Our method generates diverse structures without further training, significantly increasing the efficiency of training.

## 2 How to generate the GFlowNets parameters as you wish

To enhance the controllability and flexibility of GFlowNet parameter generation, we introduce **GenFlowNet**, a novel framework that leverages a conditional diffusion model for parameter synthesis. The preliminaries of GFlowNets and conditional diffusion models are detailed in Sec. 2.1. Subsequently, we present two core components of our framework: a parameter autoencoder (Sec. 2.2) and conditional parameter generation (Sec. 2.3). The autoencoder is designed to capture latent representations of GFlowNet parameters, which are then decoded to reconstruct the original parameters. Following this, a conditional latent diffusion model is trained to generate parameter latent representations conditioned on the structural characteristics of GFlowNets. An overview of our framework is illustrated in Fig. 2.

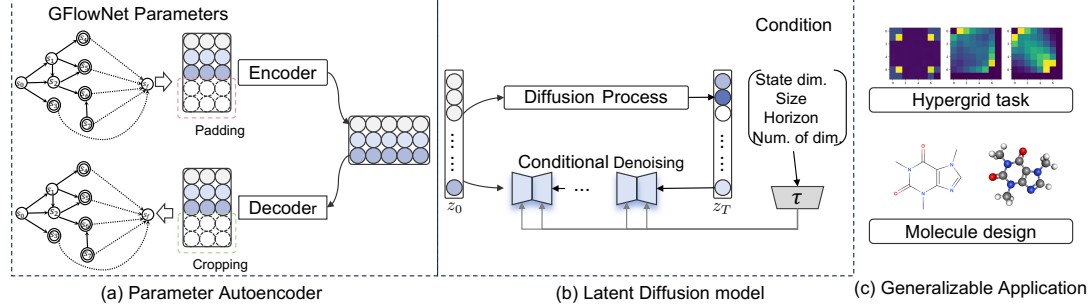

Figure 2: **Overview of the proposed GenFlowNet.** The autoencoder within our framework is utilized to extract the latent representation of GFlowNet parameters. The conditional parameter diffusion model is designed to synthesize high-performance parameters based on specific structure conditions.

## 2.1 PRELIMINARIES

**Generative Flow Networks (GFlowNets)** are designed to learn generative models over complex distributions that are defined by unnormalized density functions within structured spaces. GFlowNets model the generation of objects $x$ within a sample space $\mathcal{X}$ as a sequential decision-making process structured as an acyclic deterministic Markov Decision Process (MDP). This MDP is defined by a set of states $\mathcal{S} \supset \mathcal{X}$ and a set of actions $\mathcal{A} \subseteq \mathcal{S} \times \mathcal{S}$. The process initiates from a designated *initial state* $s_0$, which has no incoming actions and concludes in a set of *terminal states* that coincide with the objects in $\mathcal{X}$. Any object $x \in \mathcal{X}$ can be generated from $s_0$ through a sequence of actions $s_0 \to s_1 \to \cdots \to s_n = x$, with each action $(s_i, s_{i+1}) \in \mathcal{A}$. These sequences, known as *complete trajectories*, are collectively denoted as $\mathcal{T}$.

A *(forward) policy* $P_F$ is defined as a collection of distributions $P_F(s'|s)$ over the subsequent states for each non-terminal state $s \in \mathcal{S} \setminus \mathcal{X}$. This policy induces a distribution over the complete trajectories $\mathcal{T}$: $P_F(\tau = (s_0 \to s_1 \to \cdots \to s_n)) = \prod_{i=0}^{n-1} P_F(s_{i+1} \mid s_i)$. The policy enables the sampling of objects in $\mathcal{X}$ by sampling a complete trajectory $\tau \sim P_F$ and returning its terminal state. This induces a marginal distribution $P_F^T$ over $\mathcal{X}$, where $P_F^T(x)$ represents the sum of $P_F(\tau)$ over all complete trajectories $\tau$ ending in $x$ (an often intractable sum). Typically, the policy model can be implemented using simple multi-layer perceptrons (MLPs) with a few layers, though other architectures could also be applicable. The objective of GFlowNet training is to estimate a parametric policy $P_F(\cdot|\cdot; \theta)$ such that the induced distribution $P_F^T$ is proportional to a non-negative given *reward function* $R : \mathcal{X} \to \mathbb{R}_{\geq 0}$, *i.e.*, $P_F^\top(x) = \frac{1}{Z}R(x)$ for $\forall x \in \mathcal{X}$, where $Z = \sum_{x \in \mathcal{X}} R(x)$ is the unknown normalization constant (partition function).

**Conditional diffusion models** extend the standard diffusion model by incorporating conditions into both the forward and reverse processes. The conditional information, defined by $c$, allows the model to generate data tailored to specific attributes or requirements.

The forward process in conditional models involves adding noise to an initial sample while conditioning on $c$. The probability of transitioning from $x_{t-1}$ to $x_t$ under condition $c$ is modeled as a Gaussian distribution $q(x_t|x_{t-1}, c) = \mathcal{N}(x_t; \sqrt{1 - \beta_t}x_{t-1}, \beta_t I)$ where $\beta_t$ are the timestep-dependent noise levels, and $I$ represents the identity matrix. The complete forward process, conditioned on $c$, is expressed as $q(x_{1:T}|x_0, c) = \prod_{t=1}^{T} q(x_t|x_{t-1}, c)$. The conditional reverse process is designed to reconstruct the original sample from its noisiest state $x_T$ conditioned on $c$. And it is formulated by $p_\theta(x_{t-1}|x_t, c) = \mathcal{N}(x_{t-1}; \mu_\theta(x_t, t, c), \Sigma_\theta(x_t, t, c))$. In this process, $\mu_\theta$ and $\Sigma_\theta$ are functions estimated by a neural network, which also processes the condition $c$, ensuring that the recovery of data respects the conditional constraints.

The training procedure involves minimizing the Kullback-Leibler(KL) divergence between the forward and reverse conditional distributions, specifically:

$$L_{dm} = \mathbb{E}_{q(x_0, c)}[D_{KL}(q(x_{t-1}|x_t, x_0, c)||p_\theta(x_{t-1}|x_t, c))]. \tag{1}$$

During inference, the model generates new samples by conditioning on $c$ and sequentially applying the learned reverse transitions from a noise distribution. This approach enables the generation of data that closely adheres to the specified conditions.

## 2.2 PARAMETER AUTOENCODER

The autoencoder (AE) in **GenFlowNet** comprises an encoder and a decoder, both of which are based on 1D Convolutional Neural Networks (CNNs). The encoder's role is to map GFlowNet parameters into a latent space, which serves to reduce the computational and memory costs of the diffusion model (Sec. 2.3). The decoder is responsible for reconstructing these latent representations back into the original parameters.

**Dataset preparation.** To collect the training data for AE, we train GFlowNet models from scratch with stochastic gradient descent (SGD) optimizer and save dense checkpoints during the final epoch, generating a series of parameter training samples. We adopt $P$ to indicate a single training sample.

**Training Procedure.** Given a parameter training sample $P$, we flatten it into a one-dimensional vector $\Theta \in \mathbb{R}^D$, where $D$ is the dimension of the subset of parameters. The encoder is then trained

to obtain its robust latent representation, which can be used to reconstruct these flattened parameters via the decoder. The encoding and decoding processes are formalized as follows:

$$Z = \text{Encoder}(\Theta + \xi_\Theta) = \underbrace{\mathcal{E}(\Theta + \xi_\Theta, \sigma)}_{\text{encoding}}; \quad \hat{\Theta} = \text{decoder}(Z + \xi_Z) = \underbrace{\mathcal{D}(Z + \xi_Z, \rho)}_{\text{Decoding}},$$

where $\mathcal{E}(\cdot, \sigma)$ and $\mathcal{D}(\cdot, \rho)$ denote the encoder and decoder parameterized by $\sigma$ and $\rho$, respectively. $Z$ represents the latent representation of the parameter matrix, and $\hat{\Theta}$ is the reconstruction of the original parameter $\Theta$. To improve the generalization and robustness of the AE, Gaussian noise $\xi$ is introduced into both the input and latent vectors during the training process. Consistent with typical AE training (Kingma & Welling, 2013), the model is trained by minimizing the mean squared error (MSE) between the original parameters $\Theta$ and the reconstructed parameters $\hat{\Theta}$, which can be formulated as:

$$L_{\text{MSE}} = |\Theta - \hat{\Theta}|^2. \tag{2}$$

This optimization procedure ensures that the AE learns to compress and reconstruct high-dimensional GFlowNet parameters, facilitating the parameter generation in the subsequent diffusion model.

## 2.3 CONDITIONAL PARAMETER GENERATION

Diffusion models can inherently model conditional distributions through inputs like text (Reed et al., 2016), semantic maps (Isola et al., 2017; Liu et al., 2019), or images (Isola et al., 2017). Inspired by these, we design a diffusion model, taking various GFlowNet structures as conditions, to facilitate GFlowNet parameter generation.

**Generating GFlowNet via generating MLP.** GFlowNet algorithms learn a policy where the probability of sampling a terminating state $s$ is proportional to $R(s)$.

*From GFlowNet to policy:* As illustrated in Fig. 3 (a), the policy generates trajectories starting in state $s_0$ by sampling actions $a_t \in A(s)$ according to $\pi(a_t|s_t) = P_F(s_{t+1}|s_t)$, where $P_F$ represents the "forward" transition probability within the constructive GFlowNet process. The learnable forward policy governs the transition and specifies the GFlowNet, which also determines the structured internal constructions as illustrated in Fig. 3 (a) to (b). Consequently, the overall GFlowNet is predominantly determined by the forward policy.

*From policy to MLP:* In terms of neural architecture, the most straightforward GFlowNet employs a neural network that is scalable depending on the available computational resources. This network outputs a stochastic policy $\pi(a_t|s_t)$, where $s_t$ represents a partially constructed object and $a_t$ denotes one of the possible actions from $s_t$. As illustrated in preliminaries and Fig. 3 from (b) to (c), a GFlowNet forward policy can be effectively learned by a simple architecture, like a multi-layer perceptron (MLP) (Bengio et al., 2021; Liu et al., 2023). At each step, the same MLP is employed to generate a stochastic output $a_t$, which subsequently results in the next state $s_{t+1} = T(s_t, a_t)$, with $T$ being dependent on the specific application.

Hence, the GFlowNet can be determined by an MLP. This allows us to utilize the structure of the MLP as a condition for our GFlowNet parameter generation process.

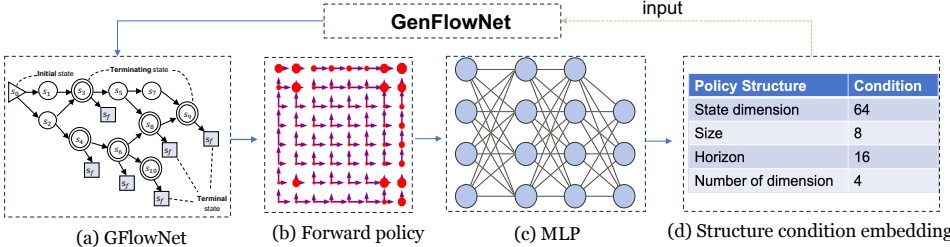

| | (a) GFlowNet | (b) Forward policy | (c) MLP | (d) Structure condition embedding |

Figure 3: **The direct modeling and conditional embedding explanation for our method.** The forward policy governs the transition of the GFlowNet and is defined as an MLP in our approach. The four structural parameters determine GFlowNet.

**Conditional embedding.** To effectively manage the diverse structural configurations of GFlowNets, we introduce an encoder $\tau_{\text{structure}}$, $(\rho; \cdot)$, where $\rho$ represents the encoder's parameters, designed to encode a GFlowNet structure into a conditional embedding. Specifically, as illustrated in Fig. 3 (c) to (d), the structure condition embedding, contains four critical structural parameters: state dimension ($s$), horizon ($h$), number of dimensions ($n$), and size ($d$). These parameters collectively define the architecture of the MLP within the GFlowNet. We concatenate these parameters to form the condition vector $c = [s, h, n, d] \in \mathbb{R}^4$. Then, this vector is fed into the structure encoder and yielding $\tau_{\text{structure}}(\rho; c)$, which will be used to control the diffusion model to generate parameters tailored to specific structure requirements in the following.

**Parameter Generation:** A straightforward strategy for generating novel parameters is by directly synthesizing them via a diffusion model. However, the computational and memory demands for this approach can become excessively high, particularly when dealing with high-dimensional parameter spaces. To mitigate this, we design a conditional diffusion model to generate latent representations of parameters tailored to the specific structural conditions $c$ of GFlowNets.

Unlike image data, which has an inherent spatial structure, GFlowNet parameters lack such spatial correlations. Therefore, instead of using 2D convolutions typically employed in image synthesis tasks, we adopt 1D convolutions to construct the diffusion model $\epsilon_\theta$, based on U-Net arachitecuture (Ronneberger et al., 2015). During the generation phase, the latent representation of parameters is progressively refined from an initial Gaussian noise distribution through successive denoising steps. Specifically, at each step $t$, the current denoised parameter latent $z_t$ and the conditioning information $c$ are passed through the diffusion model to predict the residual noise. To handle different GFlowNets and tasks, the conditional embedding $\tau_{\text{structure}}(\rho; c)$ is integrated into the intermediate features of the diffusion model $\epsilon_\theta$ via element-wise addition. The diffusion model is trained using the following objective function:

$$L_{LDM} := \mathbb{E}_{\epsilon \sim \mathcal{N}(0,1),t}[|\epsilon - \epsilon_\theta(z_t, t, \tau_{\text{structure}}(\rho; c))|^2], \tag{3}$$

where $t$ is uniformly sampled from the interval $[1, T]$, $\epsilon_\theta$ represents the denoising network parameterized by $\theta$. This formulation ensures that the model effectively learns to generate parameter latent that adhere to specific structural constraints.

**Inference procedure.** During generation, we can feed random Gaussian noise, along with the specific condition $c$ corresponding to a given structure into the diffusion model. It reverses the diffusion process and progressively refines the initial noise based on the provided condition, ultimately synthesizing a new set of parameter latent representations. Then, the pre-trained AE decoder projects it from the latent space into new parameters, which are tailored for the GFlowNet as you wish.

## 3 EXPERIMENTS

In this section, we first introduce the experimental setup. Then, we present the evaluation results, ablation study, and analysis of GenFlowNet. In our experiments, GenFlowNet effectively generates high-performance parameters for both "known" and "unknown" GFlowNets, even "unknown" tasks. This capability enables flexible and controllable synthesis of GFlowNet parameters, allowing us to obtain new high-performance GFlowNets without additional training.

### 3.1 DATASET AND IMPLEMENTATION DETAILS

**Datasets.** We evaluate our approach on a diverse range of datasets, including both training and testing sets. The training dataset comprises four distinct GFlowNet structures, each corresponding to a unique set of structural tuples. Specifically, the four training structures are denoted as Structures $A, B, C$, and $D$. For example, **Structure** $A$ is defined as [64, 16, 4, 8], where state dimension $= 64$, horizon $= 16$, number of dimensions $= 4$, and size $= 8$. The remaining structures are defined as follows: **Structure** $B$: [36, 6, 6, 6]; **Structure** $C$: [25, 5, 5, 5]; and **Structure** $D$: [36, 9, 4, 6].

In parallel with the training dataset, the test dataset is designed with four distinct structures to evaluate generalization performance. The test dataset structures can be varied to any structural configurations as required. For our experimental setup, the following structures are used in the test dataset:

**Structure** $A^*$: [9, 3, 3, 3]; **Structure** $B^*$: [16, 4, 4, 4]; **Structure** $C^*$: [36, 12, 3, 6]; and **Structure** $D^*$: [49, 7, 7, 7]. We present results of more unknown structures in Appendix 9.

**Implementation Details.** Both the autoencoder and the latent diffusion model consist of 4-layer 1D convolutional neural networks (CNNs) for the encoder and decoder. We collected a total of 200 training samples for all architectures mentioned above. We utilize **MLP-3**, a multi-layer perceptron with three linear layers and LeakyReLU activation functions, as the MLP within GFlowNet. Each of these architectures is trained from scratch independently. More details are in Appendix A.1.

### 3.2 EXPERIMENT RESULTS FOR KNOWN GFLOWNET STRUCTURE

"Known" structures refer to those that the model has encountered during training. Our method is evaluated on the benchmark hyper-grid exploration task as introduced by Bengio et al. (2021). For the sake of clarity, the main experimental results are presented using Structure $A$, with more results provided in the appendix A.2. In this task, an agent navigates a grid-like environment, starting from a corner and exploring the landscape defined by the reward function $R(x) = R_0 + R_1 \prod_i \mathbb{I}(0.25 < |x_i/H - 0.5|) + R_2 \prod_i \mathbb{I}(0.3 < |x_i/H - 0.5| < 0.4)$.

**High accuracy and low divergence.** We evaluate the quality of the generated parameters by measuring the Jensen-Shannon (JS) divergence, KL divergence, and empirical L1 loss between the ground truth probability distribution $p(x)$ and the learned probability distribution $p_\theta(x)$. The results, as shown in Tab. 1 indicate that GenFlowNet consistently matches or exceeds the baseline. In this context, the "close set" refers to GFlowNets that are present in the training set. This demonstrates the model's ability to effectively learn the distribution of high-performing parameters, including new neural field representations such as MLP parameters, achieving strong performance on various GFlowNet structures.

| | JS divergence ↓ | | KL divergence ↓ | | Empirical L1 loss↓ | | Time usage (s)↓ | |
|---|---|---|---|---|---|---|---|---|
| | GenFlowNet | Baseline | GenFlowNet | Baseline | GenFlowNet | Baseline | GenFlowNet | Baseline |
| Structure $A$ | 0.675↑ 0.005 | **0.670** | 7.276↑ 0.017 | **7.259** | 3.099e-05↑ 0.009e-05 | **3.090e-05** | **63**↓ 463 | 526 |
| Structure $B$ | **0.685**↓ 0.000 | 0.685 | **7.945**↓ 0.000 | 7.945 | **5.805e-05**↓ 0.000 | 5.805e-05 | **43**↓ 525 | 568 |
| Structure $C$ | **0.644**↓ 0.000 | 0.644 | **10.422**↓ 0.000 | 10.422 | **0.001**↓ 0.000 | 0.001 | **25**↓ 480 | 505 |
| Structure $D$ | 0.637↑ 0.003 | **0.634** | **9.467**↓ 0.000 | 9.467 | **3.000e-04**↓ 0.000 | 3.000e-04 | **60**↓ 402 | 462 |

Table 1: **Results of GenFlowNet on known GFlowNet structure .** Our approach achieved high accuracy, low divergence, and efficient time usage compared to traditional training. "Baseline" means the GFlowNet with original training.

**High quality.** Additionally, we report the average reward during training in Fig. 4a and Fig. 4b. As depicted in these figures, GFlowNets utilizing parameters generated by GenFlowNet consistently outperform the original GFlowNets, particularly during the initialization phase, indicating that more accurate flow probabilities lead to higher rewards. Beyond the superior performance in sampling distribution accuracy, as evidenced by the quantitative metrics, GenFlowNet also demonstrates strong performance in diversity evaluation, with significant improvements in mean reward. These results confirm that GenFlowNet is capable of generating high-performance model parameters tailored to specific conditions, achieving generalization across diverse known GFlowNet structure scenarios.

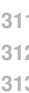 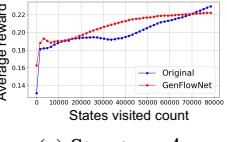 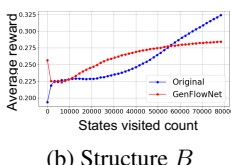 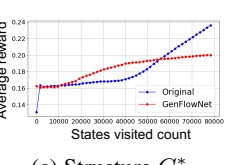 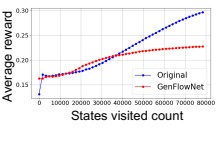

(a) Structure $A$      (b) Structure $B$      (c) Structure $C^*$      (d) Structure $D^*$

Figure 4: **Average reward of GFlowNet across structures.** Relationship between average reward and state visit counts for (a) Structure $A$ and (b) Structure $B$ in the known dataset, and (c) Structure $C^*$ and (d) Structure $D^*$ in the unknown dataset.

**Diversity in a generation.** From an alternative perspective, the models generated by GenFlowNet exhibit both diverse similarities and superior performance compared to the original models. This is illustrated in Fig. 5, which displays the parameter matrices of the generated models. Specifically, although the four models presented are representative samples derived from a set of 100 models

with identical structures (*i.e.*, Structure $A$), the weight matrices within the same layer demonstrate significant variations. This inherent diversity in the generated parameters enhances downstream applications by providing a range of robust and high-performing model configurations.

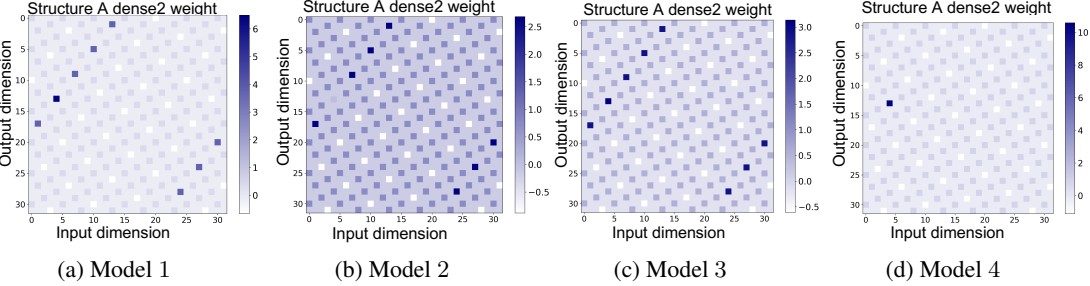

| (a) Model 1 | (b) Model 2 | (c) Model 3 | (d) Model 4 |

Figure 5: **Diversity of our generation results.** The visualization of Structure $A$ GFlowNet offers a variety of *dense2* layer's weights.

### 3.3 GENERALIZATION TO UNKNOWN GFLOWNET STRUCTURE

In addition to the known GFlowNet structure, we evaluate our model in the unknown scenario as well, where the "unknown" pertains to GFlowNet structures that have not been encountered during training. From Tab. 2, Fig. 4c, and Fig. 4d, we observe that even with unknown structures, the generated model performs well and is comparable to the model obtained through training. The verifies **the generalization capability of our method in unknown structures**. For scenarios that require a trade-off between performance and computational cost, GenFlowNet can serve as an effective initialization or pre-training method.

| | JS divergence ↓ | | KL divergence ↓ | | Empirical L1 loss↓ | | Time usage (s)↓ | |
|---|---|---|---|---|---|---|---|---|
| | GenFlowNet | Baseline | GenFlowNet | Baseline | GenFlowNet | Baseline | GenFlowNet | Baseline |
| Structure $A^*$ | **0.452**↓ 0.000 | 0.452 | 14.350↑ 0.001 | **14.349** | **0.083**↓ 0.000 | 0.083 | **13**↓ 510 | 523 |
| Structure $B^*$ | **0.479**↓ 0.002 | 0.481 | **11.022**↓ 0.003 | 11.025 | **0.009**↓ 0.000 | 0.009 | **18**↓ 576 | 594 |
| Structure $C^*$ | 0.495↑ 0.013 | **0.482** | 9.098↑ 0.282 | **8.816** | **0.001**↓ 0.000 | 0.001 | **34**↓ 437 | 471 |
| Structure $D^*$ | **0.689**↓ 0.000 | 0.689 | **5.109**↓ 0.000 | 5.109 | **3.297e-06**↓ 0.000 | 3.297e-06 | **63**↓ 533 | 596 |

Table 2: **Results of GenFlowNet on unknown GFlowNet structures.** Our approach achieved high accuracy, low divergence, and efficient time usage compared to traditional training. "Baseline" means the GFlowNet with original training.

### 3.4 GENERALIZATION TO BIOLOGICAL APPLICATION

In addition to the single task and the generalization across the known and unknown structures, we generalize and evaluate our method on a new task, which is a realistic molecule synthesis setting. The goal is to synthesize diverse molecules with desired chemical properties. We formulate molecular generation as a sequential decision process and implement it using a GFlowNet. Each state denotes a molecule graph structure, and the action space is a vocabulary of building blocks specified by junction tree modeling (Jin et al., 2018). We follow the experimental setups including the reward specification and episode constraints in (Bengio et al., 2021). The modules with the same MLP structure in the original model were replaced with the one by GenFlowNet and the performance was evaluated in Tab. 3, which shows that GenFlowNet is **training free and with efficient loss convergence in a new task.** Between the GenFlowNet and original-trained GFlowNet, in the limited iterations, the pre-trained model had more modes found and achieved a higher top-100 reward.

This means that before training for downstream tasks such as molecule generation on large network architectures, it is possible to quickly predict which GFlowNet structures are likely to perform better, allowing for more targeted and efficient training.

### 3.5 ABLATION STUDY

Extensive ablation studies are conducted in this section to highlight the characteristics of our proposed method. We focus on evaluating the performance of the generated GFlowNet parameters on

| | Training loss ↓ | | Test loss ↓ | | Top 100 reward↑ | | Modes found (R >7.5)↑ | |
|---|---|---|---|---|---|---|---|---|
| | GenFlowNet | Baseline | GenFlowNet | Baseline | GenFlowNet | Baseline | GenFlowNet | Baseline |
| I = 25,000 | **1.461**↓ 0.036 | 1.534 | **1.574**↓ 0.017 | 1.610 | **1.268**↑ 0.001 | 1.267 | **0**↑ 0 | 0 |
| I = 100,000 | **0.883**↓ 0.180 | 1.063 | 1.492↑ 0.038 | **1.454** | **2.674**↑ 0.684 | 1.990 | **2**↑ 1 | 1 |
| I = 150,000 | **0.762**↓ 0.177 | 0.939 | **1.547**↓ 0.011 | 1.558 | **2.833**↑ 0.030 | 2.803 | **2**↑ 0 | 2 |
| I = 200,000 | **0.721**↓ 0.084 | 0.805 | **1.537**↓ 0.076 | 1.613 | **3.567**↑ 0.315 | 3.252 | **3**↑ 0 | 3 |

Table 3: **Results on GenFlowNet parameters in molecule design task**. GenFlowNet demonstrates efficient and rapid convergence in molecule generation, highlighting its strong generalization capabilities and performance comparable to traditional training GFlowNet. "Baseline" means the GFlowNet with original training. "I" means the iteration number of training.

Table 4: **Ablation results of structure number $N$ in training dataset.** For Sturcture $A$ and Sturcutre $A^*$, where is written as $A$ and $A^*$ in short, larger $N$ can enhance performances.

| Structure Num. | JS divergence ↓ | | KL divergence ↓ | | Empirical L1 loss ↓ | | Time usage (s) ↓ | |
|---|---|---|---|---|---|---|---|---|
| | $A$ | $A^*$ | $A$ | $A^*$ | $A$ | $A^*$ | $A$ | $A^*$ |
| $N = 2$ | 0.675 | 0.452 | 7.277 | 14.351 | 3.099e-05 | 0.083 | 39 | 12 |
| $N = 3$ | 0.675 | 0.452 | 7.276 | 14.350 | 3.099e-05 | 0.083 | 63 | 13 |
| $N = 4$ | 0.675 | 0.453 | 7.276 | 14.351 | 3.099e-05 | 0.083 | 65 | 12 |
| $N = 5$ | 0.675 | 0.452 | 7.275 | 14.350 | 3.099e-05 | 0.083 | 63 | 11 |

Table 5: **Ablation results of model number $M$ in training dataset.** For Sturcture $A$ and Sturcutre $A^*$, where is written as $A$ and $A^*$ in short, larger $M$ can enhance performances.

| Model Num. | JS divergence ↓ | | KL divergence ↓ | | Empirical L1 loss ↓ | | Time usage (s) ↓ | |
|---|---|---|---|---|---|---|---|---|
| | $A$ | $A^*$ | $A$ | $A^*$ | $A$ | $A^*$ | $A$ | $A^*$ |
| $M = 50$ | 0.675 | 0.453 | 7.277 | 14.351 | 3.099e-05 | 0.083 | 69 | 11 |
| $M = 100$ | 0.675 | 0.453 | 7.277 | 14.351 | 3.099e-05 | 0.083 | 59 | 12 |
| $M = 200$ | 0.675 | 0.452 | 7.276 | 14.350 | 3.099e-05 | 0.083 | 60 | 13 |
| $M = 300$ | 0.675 | 0.453 | 7.275 | 14.351 | 3.098e-05 | 0.083 | 64 | 12 |

the hyper-grid exploration task. The training setup mirrors the conditions used in the experiments outlined in Tab. 1 and Tab. 2.

**Effect of structure number:** As detailed in Sec. 3.1, we utilize four distinct structures in the training and test datasets. In Tab. 4, we examine the relationship between the number of structures, $N$, and performance, using a fixed set of 200 models for training. The results indicate that varying the number of structures has minimal impact on the performance of the parameter generator.

**Number of training models:** Tab. 5 varies the size of training data, *i.e.* the number of original models, with a fixed number of training structures as 3. We find the performance gap of best results in different numbers of the original models is minor.

**Generalization of model size:** As illustrated in both Tab. 4 and Tab. 5, it is usual that the training datasets consist of only GFlowNets with small network structures and test to generate the new GFlowNet with larger structures. This demonstrates the generalization of the model size for the generator. GFlowNets with smaller network structures as the training dataset, while generating GFlowNets for larger structures, can demonstrate that the resulting performance is also comparable.

## 3.6 ANALYSIS

**MLP parameter-space diffusion:** We explain the reason why we can use the diffusion model to generate MLP parameters well, even in unknown scenarios. Since we consider each set of MLP parameters (weights and biases) $S$ as a flattened 1D vector for diffusion, it enables a general formulation for modeling neural fields, as the MLP parameters are agnostic to varying-dimensional data. This makes GenFlowNet flexible to a variety of neural field representations; in particular, it generalizes to different GFlowNet structures from the training dataset. We also observe as in Tab. 6 that the variance across structures and models varies a little, which means that the neural field is compact enough to be demonstrated by generative modeling of MLP representation with our diffusion model and each layer parameters matter similarly for the final output representation. This is also observed in (Erkoç et al., 2023). The appearance of the generated parameters is similar to the baseline, which verifies the robustness of our method.

Table 6: **Test parameter variance across layers & models**. It evaluates that each layer in different GFlowNets matters. The variance of parameters across multiple GFlowNet structures equals the expected variance of the individual models for unknown GFlowNet structures generated from the GenFlowNet. "Baseline" means the GFlowNet with original training.

| Parameter | Structure $A^*$ (50 models) | | Structure $B^*$ (50 models) | | Structure $B^*$ (100 models) | | Structure $C^*$ (100 models) | |
|---|---|---|---|---|---|---|---|---|
| | GenFlowNet | Baseline | GenFlowNet | Baseline | GenFlowNet | Baseline | GenFlowNet | Baseline |
| dense1.weight | 0.005 | 0.005 | 0.009 | 0.010 | 0.009 | 0.009 | 0.014 | 0.013 |
| dense1.bias | 0.007 | 0.004 | 0.013 | 0.008 | 0.011 | 0.008 | 0.010 | 0.016 |
| dense2.weight | 0.001 | 0.010 | 0.011 | 0.010 | 0.010 | 0.010 | 0.010 | 0.010 |
| dense2.bias | 0.012 | 0.009 | 0.012 | 0.011 | 0.012 | 0.009 | 0.010 | 0.014 |
| dense3.weight | 0.012 | 0.012 | 0.011 | 0.011 | 0.010 | 0.010 | 0.009 | 0.010 |
| dense3.bias | 0.002 | 0.002 | 0.012 | 0.014 | 0.014 | 0.007 | 0.001 | 0.013 |

**Difference with vision model parameter generation:** We investigate several factors contributing to the limited generalizability observed in vision model parameter generation, as noted in prior work (Wang et al., 2024; Jin et al., 2024). However, our analysis demonstrates that GFlowNet, when applied across different network architectures, achieves substantial improvements in generalization performance. Firstly, in our parameter generation, the diffusion process leverages a more direct form of conditioning: the latent diffusion model is conditioned on a set of parameters that explicitly define the MLP's architecture.

Secondly, in addition to the explicit explanation of the condition itself, the MLP has an advantage in terms of parameter efficiency as in Tab. 7. The generalization of model size for the GenFlowNet is further demonstrated by embedding a GFlowNet, trained on the dataset with a smaller network structure, into a larger network structure during generalization evaluation in Sec. 3.4. The model parameter count reflects this scalability, highlighting the framework's ability to adapt effectively across GFlowNets with different network structure sizes.

Thirdly, another contributing factor is the structural similarity of the network architecture. To understand neural networks, researchers often

Table 7: **Parameter amount of different networks**. It shows the efficient parameter amounts of GFlowNet. Model $A$ and model $B$ refer to GFlowNet for molecule design and hypergrid task, ResNet 18 and ResNet 34, Transformer with 4 and 8 heads respectively.

| Model | Total number↓ | |
|---|---|---|
| | Model $A$ | Model $B$ |
| MLP | $193,709$ | $2,339$ |
| ResNet | $11,689,512$ | $21,797,672$ |
| Transformer | $3,952,133$ | $12,625,420$ |

use similarity metrics to measure how similar or different two neural networks are to each other. One of the metrics called centered kernel alignment (CKA)(Kornblith et al., 2019) can reveal pathology in neural network representations and is equally effective at revealing relationships between layers of different architectures. Fig. 6 shows the relationship between different layers of GFlowNets, transformers, and ResNets with two structures. The GFlowNet representations seem to be "similar" on average and have lower variance, which can be compact for our model to learn the neural field information from its parameters.

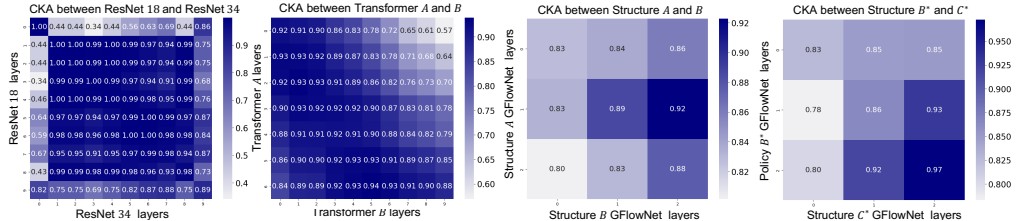

(a) Vision Tasks: ResNet vs. ResNet and Transformer vs. Transformer

(b) GFlowNet Tasks: Structure $A$ vs. Structure $B$ and Structure $B^*$ vs. Structure $C^*$

Figure 6: **Comparison of CKA performance**: (a) Vision Tasks (ResNet and Transformer), (b) GFlowNet. The visualization of the CKA demonstrates the low layer variance of GFlowNet, which is evidence of the generalization. Additionally, ResNet 18 and ResNet 34 take the average pooling.

## 4 RELATED WORK

**Diffusion Models:** Diffusion models have achieved outstanding results in visual generation tasks. These methods (Ho et al., 2020; Dhariwal & Nichol, 2021; Ho et al., 2022; Peebles & Xie, 2023; Hertz et al., 2022; Li et al., 2023) are grounded in non-equilibrium thermodynamics (Jarzynski, 1997; Sohl-Dickstein et al., 2015), following a generative process similar to GANs (Zhu et al., 2017; Isola et al., 2017; Brock et al., 2018), VAEs (Kingma & Welling, 2013; Razavi et al., 2019), and flow-based models (Dinh et al., 2014; Rezende & Mohamed, 2015). Broadly, diffusion models can be classified into three main branches. The first focuses on improving synthesis quality, as demonstrated by models such as DALL·E 2 (Ramesh et al., 2022), Imagen (Saharia et al., 2022), and Stable Diffusion (Rombach et al., 2022). The second branch aims to accelerate sampling speeds, featuring models like DDIM (Song et al., 2020), Analytic-DPM (Bao et al., 2022), and DPM-Solver (Lu et al., 2022). The final branch reinterprets diffusion models from a continuous perspective, exemplified by score-based models (Song & Ermon, 2019; Feng et al., 2023).

**Conditional Generation:** Conditional generation has garnered significant attention in both computer vision and natural language processing. Three key frameworks dominate this area: conditional GANs (Mirza & Osindero, 2014; Isola et al., 2017; Zhu et al., 2017), conditional VAEs (Sohn et al., 2015; Yan et al., 2016), and conditional diffusion models (Rombach et al., 2022), all of which integrate conditions to guide the generative process. This approach facilitates the creation of visually coherent and semantically meaningful data samples. Conditional GANs incorporate attribute or label information to condition the generation process, while conditional diffusion models advance this further by generating high-quality images from textual descriptions. These diffusion-based models have outperformed GANs in terms of both visual coherence and semantic accuracy. Building on the success of conditional diffusion models, we extend this concept to the generation of neural network parameters, conditioned on specific attributes, for efficient model tuning.

**Parameter Generation:** Parameter generation has seen rapid advancements, with approaches like HyperNetworks (Ha et al., 2016) and generative models of neural network checkpoints (Peebles et al., 2022) emerging as notable contributions. Ha et al. (2016) introduced HyperNetworks, which generates neural network parameters by learning from an auxiliary network. Finn et al. (2017) proposed Model-Agnostic Meta-Learning (MAML), a technique that learns an initialization that allows efficient fine-tuning across tasks. More recently, Peebles et al. (2022) developed G.pt, a model designed to predict parameter updates given an initial parameter vector and a prompted loss function. Schürholt et al. (2022) trained an autoencoder on a model zoo to learn hyper-representations for generating new model weights, while Knyazev et al. (2021) applied a GNN-based model to sample network parameters. In a similar vein, Erkoç et al. (2023) generated neural implicit fields by leveraging synthesized MLP weights. Wang et al. (2024) extended diffusion models to generate high-performing neural network parameters across various architectures and datasets. Unlike these prior works, our approach focuses on conditional and controllable parameter generation, aiming to produce high-performance parameters tailored to specific GFlowNet. Our method not only achieves high-performance parameter generation but also demonstrates strong generalizability across different tasks and architectures.

## 5 DISCUSSION AND CONCLUSION

In this work, we propose **GenFlowNet**, a framework for generating GFlowNet parameters based on a given structure. Our approach combines an autoencoder with a conditional latent diffusion model to capture the distribution of GFlowNet parameters, allowing us to obtain high-performing parameters without further training. Our experiments demonstrate that GenFlowNet effectively generates novel, high-quality, and diverse parameters and exhibits remarkable generalizability across diverse tasks and structures.

**Limitations and future works.** However, there are still unresolved challenges, including better GFlowNet structure design in various tasks, and ensuring the performance stability of the generated parameters across more applications. Furthermore, integrating scientific knowledge into conditions offers promising directions for GenFlowNet.

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

## A    APPENDIX

**Experimental settings:**

- Sec. A.1: Training details of GenFlowNet and the model details.

**Additional results:**

- Sec. A.2: The detailed visualization and results for GenFlowNet.

## A.1 EXPERIMENT SETUP

In this section, we show detailed experiment setups, including dataset information and training configuration.

**Training recipe:** We provide our basic training recipe with specific details as follows. We introduce these details of general training hyperparameters, autoencoder, and conditional latent diffusion model, respectively. It may be necessary to make adjustments to the learning rate and the training iterations for different tasks.

| Training Setting | Configuration |
|---|---|
| $K$, *i.e.*, the number of original models | 200 |
| batch size | 200 |
| Autoencoder | |
| optimizer | AdamW |
| learning rate | 1e-3 |
| training iterations | 30,000 |
| optimizer momentum | betas = (0.9, 0.999) |
| weight decay | 2e-6 |
| $\xi_\Theta$, *i.e.*, noise added on the input parameters | 0.001 |
| $\xi_Z$, *i.e.*, noise added on the latent representations | 0.1 |
| Diffusion | |
| optimizer | AdamW |
| learning rate | 1e-3 |
| training iterations | 30,000 |
| optimizer momentum | betas = (0.9, 0.999) |
| weight decay | 2e-6 |
| ema $\beta$ | 0.9999 |
| betas start | 1e-4 |
| betas end | 2e-2 |
| betas schedule | linear |
| $T$, *i.e.*, maximum time steps in the training stage | 1000 |

Table 8: Our basic training recipe

All the experiments are conducted on an 8-V100 GPU.

**Preparation for Training Dataset:** In the training process of the AE component, we utilized 200 models for each structure of the GFlowNet, with these models serving as the samples. With a total of 4 structures included in the training dataset, this resulted in 800 samples overall. The dataset used for training and evaluation was the hypergrid task, which was specifically chosen to align with the requirements of our approach.

**Training of Autoencoder and Conditional Parameter Diffusion:** We introduce details of the training process of the autoencoder and the diffusion models. For the autoencoder, we use the parameter data to train the autoencoder to encode the GFlowNet parameters into a 256-dimensional latent space. For the conditional diffusion model, we use a network structure condition extractor to extract the MLP structure features of the GFlowNet and merge the features into the diffusion model as condition information.

## A.2 ADDITIONAL EXPERIMENT RESULTS

**Additional results of GenFlowNet on unknown GFlowNet structures.** For our experimental setup, the following structures are used in the additional test dataset: **Structure** $E^*$: [4, 2, 2, 2]; **Structure** $F^*$: [16, 8, 2, 4]; **Structure** $G^*$: [81, 27, 3, 9]; and **Structure** $H^*$: [100, 25, 4, 10].

**The visualization of Structure** $A$ **GFlowNet different layer weights**

- Dense layer 1
- Dense layer 3

| | JS divergence ↓ | | | KL divergence ↓ | | | Empirical L1 loss↓ | | | Time usage (s)↓ | |
|---|---|---|---|---|---|---|---|---|---|---|---|
| | GenFlowNet | Baseline | | GenFlowNet | Baseline | | GenFlowNet | Baseline | | GenFlowNet | Baseline |
| Structure $E^*$ | **0.001**↓ 0.000 | 0.001 | | 0.002↑ 0.001 | **0.001** | | **0.020**↓ 0.000 | 0.020 | | **13**↓ 510 | 488 |
| Structure $F^*$ | **0.424**↓ 0.003 | 0.427 | | **2.391**↓ 0.001 | 2.390 | | 0.025↑ 0.003 | **0.022** | | **19**↓ 549 | 568 |
| Structure $G^*$ | 0.625↑ 0.001 | **0.624** | | **8.244**↓ 0.000 | 8.244 | | **1.000e-04**↓ 0.000 | 1.000e-04 | | **43**↓ 586 | 629 |
| Structure $H^*$ | **0.685**↓ 0.000 | 0.685 | | **5.544**↓ 0.000 | 5.544 | | **5.000e-05**↓ 0.000 | 5.000e-05 | | **54**↓ 643 | 697 |

Table 9: **Results of GenFlowNet on unknown GFlowNet structures.** Our approach achieved high accuracy, low divergence, and efficient time usage compared to traditional training. "Baseline" means the GFlowNet with original training.

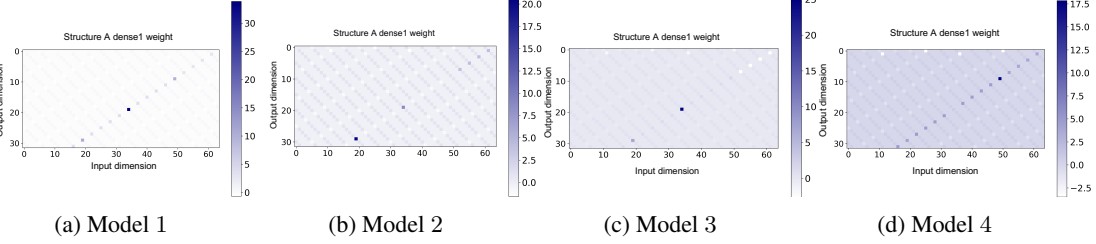

(a) Model 1      (b) Model 2      (c) Model 3      (d) Model 4

Figure 7: **Diversity of our generation results.** The visualization of Structure $A$ GFlowNet offers a variety of *dense1* layer's weights.

**Ablation results of structure number $N$ in training dataset.**

- Structure $B$ and Structure $B^*$

Table 10: **Ablation results of structure number $N$ in training dataset.** For Structure $B$ and Structure $B^*$, where is written as $B$ and $B^*$ in short, larger $N$ can enhance performances.

| Structure Num. | JS divergence ↓ | | KL divergence ↓ | | Empirical L1 loss ↓ | | Time usage (s) ↓ | |
|---|---|---|---|---|---|---|---|---|
| | $B$ | $B^*$ | $B$ | $B^*$ | $B$ | $B^*$ | $B$ | $B^*$ |
| $N = 2$ | 0.685 | 0.479 | 7.948 | 11.022 | 5.805e-05 | 0.009 | 39 | 18 |
| $N = 3$ | 0.685 | 0.479 | 7.946 | 11.022 | 5.805e-05 | 0.009 | 43 | 18 |
| $N = 4$ | 0.685 | 0.479 | 7.947 | 11.021 | 5.805e-05 | 0.009 | 41 | 19 |
| $N = 5$ | 0.685 | 0.479 | 7.946 | 11.021 | 5.805e-05 | 0.009 | 45 | 19 |

- Structure $C$ and Structure $C^*$
- Structure $D$ and Structure $D^*$

**Ablation results of model number $M$ in training dataset.**

- Structure $B$ and Structure $B^*$
- Structure $C$ and Structure $C^*$
- Structure $D$ and Structure $D^*$

**Visualization of hypergrid task**

- Training for structure $B$

**The uncertainty evaluation of GenFlowNet.** We list the results for uncertainty evaluation in the training dataset in Table 16. The results are presented in terms of best, average, and median performance.

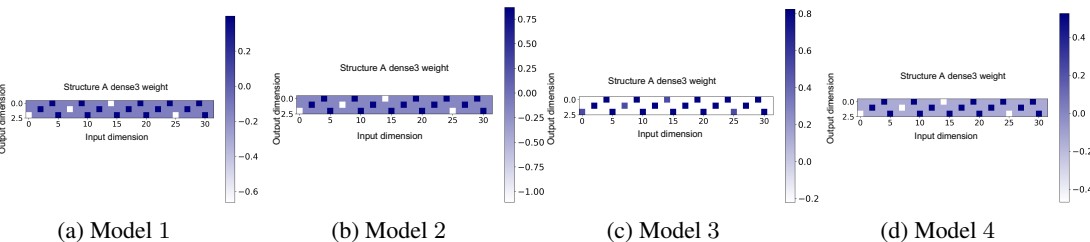

|     (a) Model 1     |     (b) Model 2     |     (c) Model 3     |     (d) Model 4     |

Figure 8: **Diversity of our generation results.** The visualization of Structure $A$ GFlowNet offers a variety of *dense3* layer's weights.

Table 11: **Ablation results of structure number $N$ in training dataset.** For Structure $C$ and Structure $C^*$, where is written as $C$ and $C^*$ in short, larger $N$ can enhance performances.

| Structure Num. | JS divergence ↓ | | KL divergence ↓ | | Empirical L1 loss ↓ | | Time usage (s) ↓ | |
|---|---|---|---|---|---|---|---|---|
|  | $C$ | $C^*$ | $C$ | $C^*$ | $C$ | $C^*$ | $C$ | $C^*$ |
| $N = 2$ | 0.644 | 0.452 | 10.422 | 14.351 | 0.001 | 0.083 | 26 | 32 |
| $N = 3$ | 0.644 | 0.452 | 10.422 | 14.350 | 0.001 | 0.083 | 25 | 34 |
| $N = 4$ | 0.644 | 0.453 | 10.421 | 14.351 | 0.001 | 0.083 | 28 | 34 |
| $N = 5$ | 0.643 | 0.452 | 10.422 | 14.350 | 0.001 | 0.083 | 38 | 35 |

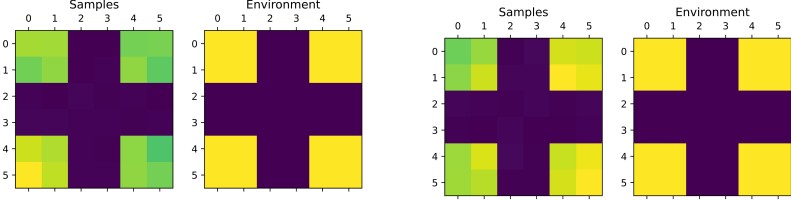

Visualization of hypergrid task by original training GFlowNet    Visualization of hypergrid task by GenFlowNet

Figure 9: The visualization of hypergrid task for Structure $B$ via original training GFlowNet(left) and GenFlowNet(right). "samples" means the generated results while "environments" means the ground truth. The color gets more closer to yellow, the more accurate the result is.

Table 12: **Ablation results of structure number $N$ in training dataset.** For Structure $D$ and Structure $D^*$, where is written as $D$ and $D^*$ in short, larger $N$ can enhance performances.

| Structure Num. | JS divergence ↓ | | KL divergence ↓ | | Empirical L1 loss ↓ | | Time usage (s) ↓ | |
|---|---|---|---|---|---|---|---|---|
|  | $D$ | $D^*$ | $D$ | $D^*$ | $D$ | $D^*$ | $D$ | $D^*$ |
| $N = 2$ | 0.638 | 0.689 | 9.467 | 5.109 | 3.000e-04 | 3.297e-06 | 56 | 63 |
| $N = 3$ | 0.637 | 0.689 | 9.467 | 5.109 | 3.000e-04 | 3.297e-06 | 60 | 63 |
| $N = 4$ | 0.637 | 0.489 | 9.467 | 5.109 | 3.000e-04 | 3.297e-06 | 58 | 62 |
| $N = 5$ | 0.637 | 0.489 | 9.466 | 5.109 | 3.000e-04 | 3.297e-06 | 61 | 63 |

Table 13: **Ablation results of model number $M$ in training dataset.** For Structure $B$ and Structure $B^*$, where is written as $B$ and $B^*$ in short, larger $M$ can enhance performances.

| Model Num. | JS divergence ↓ | | KL divergence ↓ | | Empirical L1 loss ↓ | | Time usage (s) ↓ | |
|---|---|---|---|---|---|---|---|---|
|  | $B$ | $B^*$ | $B$ | $B^*$ | $B$ | $B^*$ | $B$ | $B^*$ |
| $M = 50$ | 0.686 | 0.479 | 7.946 | 11.022 | 5.805e-05 | 0.009 | 43 | 25 |
| $M = 100$ | 0.685 | 0.479 | 7.946 | 11.022 | 5.805e-05 | 0.009 | 42 | 25 |
| $M = 200$ | 0.685 | 0.479 | 7.945 | 11.022 | 5.805e-05 | 0.009 | 43 | 30 |
| $M = 300$ | 0.685 | 0.479 | 7.945 | 11.022 | 5.805e-05 | 0.009 | 44 | 28 |

Table 14: **Ablation results of model number $M$ in training dataset.** For Structure $C$ and Structure $C^*$, where is written as $C$ and $C^*$ in short, larger $M$ can enhance performances.

| Model Num. | JS divergence ↓ | | KL divergence ↓ | | Empirical L1 loss ↓ | | Time usage (s) ↓ | |
|---|---|---|---|---|---|---|---|---|
|  | $C$ | $C^*$ | $C$ | $C^*$ | $C$ | $C^*$ | $C$ | $C^*$ |
| $M = 50$ | 0.644 | 0.453 | 10.422 | 14.351 | 0.001 | 0.083 | 26 | 32 |
| $M = 100$ | 0.644 | 0.453 | 10.422 | 14.351 | 0.001 | 0.083 | 25 | 32 |
| $M = 200$ | 0.644 | 0.452 | 10.422 | 14.350 | 0.001 | 0.083 | 25 | 34 |
| $M = 300$ | 0.644 | 0.453 | 10.422 | 14.351 | 0.001 | 0.083 | 25 | 33 |

Table 15: **Ablation results of model number $M$ in training dataset.** For Structure $D$ and Structure $D^*$, where is written as $D$ and $D^*$ in short, larger $M$ can enhance performances.

| Model Num. | JS divergence ↓ | | KL divergence ↓ | | Empirical L1 loss ↓ | | Time usage (s) ↓ | |
|---|---|---|---|---|---|---|---|---|
| | $D$ | $D^*$ | $D$ | $D^*$ | $D$ | $D^*$ | $D$ | $D^*$ |
| $M = 50$ | 0.637 | 0.453 | 9.468 | 14.351 | 3.000e-04 | 3.297e-06 | 69 | 60 |
| $M = 100$ | 0.637 | 0.689 | 9.468 | 5.109 | 3.000e-04 | 3.297e-06 | 59 | 58 |
| $M = 200$ | 0.637 | 0.689 | 9.467 | 5.109 | 3.000e-04 | 3.297e-06 | 60 | 63 |
| $M = 300$ | 0.636 | 0.453 | 9.467 | 14.351 | 3.000e-04 | 3.297e-06 | 57 | 63 |

| Structure | JS Divergence ↓ | KL Divergence ↓ | Empirical L1 Loss ↓ |
|---|---|---|---|
| Structure $A$ | 0.674/0.675/0.677 | 7.275/7.276/7.275 | 3.097e-05/3.099e-05/3.099e-05 |
| Structure $B$ | 0.685/0.685/0.686 | 7.942/7.945/7.943 | 5.803e-06/5.805e-05/5.804e-05 |
| Structure $C$ | 0.641/0.644/0.643 | 10.421/10.422/10.422 | 0.001/0.001/0.001 |
| Structure $D$ | 0.636/0.637/0.637 | 9.463/9.467/9.466 | 3.000e-04/3.000e-04/3.000e-04 |

Table 16: Uncertainty measure on Divergence Metrics and Empirical L1 Loss

