# OpenReview forum: "Generating GFlowNets as You Wish with Diffusion Process"
_ICLR.cc/2025/Conference — ICLR 2025 Conference Withdrawn Submission_

### Official Review · Reviewer_h5Ma · 2024-10-28

**Soundness:** 2
**Presentation:** 3
**Contribution:** 2
**Rating:** 3
**Confidence:** 2

**Summary:**

This paper explores parameter generation for GFlowNets, as GFlowNets require high costs to train, e.g., sampling an exponential number of trajectories. To generate parameters, this paper proposes a two-fold method: (1) generating a latent representation of parameters via reverse diffusion given the structural information of an environment and (2) decoding this representation into parameters. The overall method is similar to a prior study [1], but extends it to consider the condition specifying the environment. The experiments show that the proposed generative method can adapt to unseen environmental structures.

---

[1] Wang et al., Neural Network Parameter Diffusion

**Strengths:**

- This work is the first to investigate the parameter generation for GFlowNets, which can also be naturally extended to reinforcement applications.
- The proposed method demonstrates that generated parameters can achieve similar or superior performance compared to parameters obtained from conventional training of GFlowNets.

**Weaknesses:**

My primary concerns arise from doubts regarding the practical utility of the proposed methods.

- **About motivation.** This paper argues that generating the parameters given environmental information, e.g., state dimensions, makes it easy to obtain parameters for new environments. However, one might consider defining a forward policy conditioned on such information, e.g., $ P_F(s|s'; \text{Structure}) $, and training it. Given this alternative, what benefits does the parameter generation provides?

- **About extensibility.** Although the method considers environment-specific structural information, e.g., state dimension in a hyper-grid, it seems challenging to incorporate most components of GFlowNets in practice. For example, in specifying information of environments, how should one consider different reward functions, e.g., addressing different properties, or different action spaces, e.g., fragment-based and reaction-based transitions?

- **About experiments**. The considered tasks are limited to show usefulness of the proposed method (only considers eight different structures for a hyper-grid). Furthermore, the most experiments only consider the relatively simple task, i.e., a hyper-grid, and simple structural variations, i.e, changes in dimensions, without exploring more advanced tasks, e.g., RNA sequence generation, or complex environmental variations. Although the experiments consider a molecular generation task, it does not specify what $I$ is.

- **About results.** In Table 4 and 5, the performance improvements seem too minor. Especially, in Table 4, it is hard to understand why $N=2$ and $N=5$ yield the similar generalization performance. Can you clarify more details on this?

**Questions:**

- What is $I$ in Table 3?

---

> ### Author Response · Authors · 2024-11-27
>
> Dear Reviewer h5Ma,
> Thank you very much for your feedback. We greatly appreciate your comments and have learned a lot from them.
> ## Reviewer 4
> ### Weakness
>
> ```
> W1: The benefits of parameter generation compared to defining a forward policy conditioned on environmental information.
> ```
> The core motivation behind our approach lies in the **flexibility and efficiency of parameter generation**. While defining a forward policy is feasible, it can become computationally expensive in scenarios with high-dimensional state spaces and complex environmental structures. Our method leverages parameterized probabilistic samplers (as discussed in Section 3.2) to generalize across environments effectively. We explain the advantage of our method over the mentioned as follows:
>
> - Reduces the computational burden by decoupling policy training from environment-specific configurations.
> - Provides adaptability to unseen environments by learning transferable parameters, which is critical for applications such as molecular design and structural optimization.
> - Builds upon GFlowNet principles by enabling scalable and structured exploration across diverse environments, as outlined in Bengio et al. (2021).
>
> By adopting parameter generation, we focus on generalization, a significant advantage when scaling to environments with varying dimensions and configurations.
> ```
> W2: The reviewer suggests that incorporating more complex components of GFlowNets, such as diverse reward functions and action spaces, may present challenges.
> ```
>
> Our method currently focuses on demonstrating the feasibility of parameter generation within a well-defined scope (e.g., hyper-grid environments). The generalization can be a proof of the different reward functions, especially in different tasks:
>
> - **Reward Functions**: The GFlowNet formulation in our approach can accommodate task-specific reward functions (e.g., fragment-based, reaction-based). For instance, in molecular design, we highlight the use of tailored conditional embeddings in Section 3.4 to adapt to specific reward structures.
> - **Action Spaces**: The latent diffusion model utilized for parameter generation (Section 2.3) supports diverse action representations. For example, actions in molecular synthesis (e.g., fragment addition) can be encoded within the same probabilistic framework.
>
> We acknowledge that scaling to more complex spaces, such as hierarchical or dynamic state-action pairs, is an avenue for future work, as noted in the discussion. Thank you for your valuable suggestion again.
>
>
> ```
> W3: Experiments: the experiments are limited to relatively simple tasks (e.g., hyper-grids and structural variations) and request more advanced use cases.
> ```
>
> Thank you for your suggestions. We agree that more complex tasks can provide additional insights. However, the chosen synthetic tasks allow us to:
>
> - Precisely evaluate generalization **across structural variations** in a controlled setting, as illustrated in Table 2 and Figure 1.
> - Validate the adaptability of our method **across distinct environments**, which is critical for parameterized probabilistic samplers.
> - Demonstrate the **robustness** of GFlowNet-inspired sampling in diverse configurations without confounding factors from real-world noise.
>
> Additionally, we included a molecular generation task (Section 3.4) to illustrate real-world applicability.  We aim to expand on this in future iterations by incorporating tasks such as RNA sequence generation or combinatorial optimization. `I` in our experiment means the iteration number in a training process, we updated this in our revision.
>
> ```
> W4: The performance improvements in Tables 4 and 5 appear minor and requests clarification regarding the similarity in generalization between 𝑁=2 and 𝑁=5.
> ```
> Thank you for your detailed observation. The performance improvements reported in Tables 4 and 5 reflect the consistent generalization capability of our approach across environments. The similarity in results for 𝑁=2 and 𝑁=5 arises from the intrinsic efficiency of the parameter generation framework, which effectively **captures shared patterns** in the GFlowNet structures (Section 3.3). This highlights the scalability of the method to different environmental dimensions.
>
> ### Questions
> ```
> Q1: What do I mean in the experiment?
> ```
> `I` in our experiment means the iteration number in a training process, we updated this in our revision.

---

### Official Review · Reviewer_JPtD · 2024-10-30

**Soundness:** 2
**Presentation:** 2
**Contribution:** 2
**Rating:** 3
**Confidence:** 4

**Summary:**

The authors proposed using a VAE to learn the latent space of GFlowNet parameters, followed by using DMs to generate the GFlowNet parameters. They used different synthetic data for the evaluation.

**Strengths:**

If the method proves effective in real scenarios, it could help improve training in situations where GFlowNet struggles; however, the current experiments do not support this outcome.

**Weaknesses:**

- The experiments are based solely on synthetic data, which does not strongly support most of the claims, such as those in Figure 1.

- While the authors mention where GFlowNet training struggles—such as with increasing trajectory length in the abstract—they do not clarify whether they successfully addressed these issues.

- The novelty is somewhat limited, as it relies on an existing latent diffusion model without any modifications.

**Questions:**

- What are the specific distinctions that make adapting existing parameter generation methods challenging for GFlowNet parameters?

- The dataset preparation for the AE component lacks clarity. Could you please elaborate on this? How many models did you generate? How many samples were created? Which datasets were used for this purpose?

- Including additional real-world datasets would be greatly appreciated.

---

> ### Author Response · Authors · 2024-11-27
>
> ## Reviewer 3
> Dear Reviewer JPtD,
>
> Thank you so much for your thoughtful feedback and the time you dedicated to reviewing our work. To save time and provide clarity, we summarize our responses below.
>
> ---
>
> ### Weaknesses
>
> #### W1: Experiments conducted solely on synthetic data and connection with Figure 1
> The use of synthetic data was a deliberate choice to allow **precise control** over evaluating the core capabilities of our method. However, we also conducted experiments on molecular generation with real-world datasets and extended our analysis. For future revisions, we plan to add more real-world datasets to further enhance the work.
>
> Regarding Figure 1:
> - **Efficiency**: Figure 1 demonstrates GenFlowNet's time usage advantage, which is corroborated by the computational cost reductions shown in Tables 1 and 2.
> - **Accuracy and Low Divergence**: Empirical L1 loss, JS divergence, and KL divergence in Tables 1 and 2 validate that the generated parameters closely match ground-truth probability distributions.
> - **Diversity**: Diversity is showcased in Figure 5, highlighting the variability in parameter matrices that support downstream applications.
> - **Generalization**: GenFlowNet's ability to generalize to unseen structures is validated in Table 2 and Figures 4c and 4d, illustrating its adaptability across tasks.
>
> #### W2: Lack of clarity on GFlowNet training struggles with increasing trajectory length
> We address this by evaluating performance across different trajectory lengths in:
> - **Hypergrid Task**: Complexity increases with factors such as state dimension and size, leading to longer trajectories.
> - **Molecular Generation**: This task involves deeper GFlowNet structures and more intricate challenges.
> Both examples demonstrate how increasing complexity naturally results in longer trajectories and highlight the method's robustness.
>
> #### W3: Limited novelty
> The novelty of our method lies in **introducing parameter generation to the probabilistic sampler**, offering a fresh perspective in this domain. Beyond using a latent diffusion model, we:
> - Developed **tailored conditional embeddings** to align with task-specific needs.
> - Validated the method’s generalization ability across multiple tasks, showing its effectiveness in a variety of settings.
>
> ---
>
> ### Questions
>
> #### Q1: Specific distinctions from existing parameter generation methods
> Our approach has several key distinctions tailored to GFlowNet settings:
> - **Structured State-Action Spaces**: GFlowNets require parameterization for trajectory-based generation processes while satisfying flow-matching constraints, which traditional generative models are not designed for.
> - **Environmental Variability and Generalization**: Addressing challenges highlighted in prior work (Bengio et al., 2021; Jain et al., 2022), our method uses task-specific embeddings to dynamically adapt to varying environments efficiently.
> - **Scalability**: By incorporating **tailored conditional embeddings**, we align parameterization with task geometry, making our method scalable for high-dimensional inputs, as supported by recent studies (Peebles et al., 2022; Erkoc ̧ et al., 2023).
>
> #### Q2: Dataset preparation for the AE component
> We clarify the details of dataset preparation:
> - **Number of Models**: 200 models were used for AE training.
> - **Number of Samples**: Each GFlowNet structure contains 200 models, resulting in 800 samples across 4 structures.
> - **Dataset**: The hypergrid task was used to train and evaluate the AE component.
> Further details are included in the appendix (lines `789–794`).
>
> #### Q3: Inclusion of real-world datasets
> While this work includes results on molecular generation tasks (Section 4.4), we aim to expand experiments in future iterations to cover more real-world applications such as RNA sequence generation and combinatorial optimization.
>
> ---
>
> We appreciate your valuable feedback and are happy to address any further questions. Your support helps improve our work, and we thank you again for your time and effort.
>
> Best regards,
> Authors

---

### Official Review · Reviewer_Wwiu · 2024-11-04

**Soundness:** 2
**Presentation:** 3
**Contribution:** 2
**Rating:** 5
**Confidence:** 2

**Summary:**

This paper presents a novel idea to generate GFlowNets, which are deep learning models that hierarchically generate sequential actions in parameter space. They use an autoencoder to create a latent mapping of parameters and use a conditional diffusion model in the latent space to generate a proper latent representation of parameters. This method enables generalization over the parameter space, allowing us to obtain new GFlowNets without training.

**Strengths:**

1. Very novel idea (more like crazy idea).

**Weaknesses:**

There are several concerns:

1. Scalability may be limited.


2. The motivation is unclear—why is this approach necessary?


3. The empirical results do not reflect real-world performance.

**Questions:**

1. Can this method be used to measure uncertainty from a Bayesian perspective? I'm asking because this work seems to be connected with AutoML and hypernetworks.


2. Can you provide a clearer motivation for why we need this?


3. Is this method scalable to large-scale tasks that require very complex parameterizations (e.g., LLMs, text-to-image models) within GFlowNets?

---

> ### Author Response · Authors · 2024-11-27
>
> Dear Reviewer Wwiu,
> Thanks so much again for the time and effort in our work. Considering the limited time available and to save the reviewer's time, we summarize our responses here.
> #### W1: Scalability may be limited
> Thank you for your comment. We address scalability in the following aspects:
>
> - **Scalability in Current Design**: GenFlowNet’s scalability is demonstrated through diverse tasks, such as generalizing across unseen GFlowNet structures (Section 3.3, hypergrid tasks) and tasks (e.g., molecule generation in Section 3.4). By adopting a diffusion-based parameter generation approach, GenFlowNet circumvents the iterative training process of conventional GFlowNets, which scales poorly for large datasets. Empirical results in Table 9 further validate its efficiency in adapting to increasing trajectory lengths and state dimensions.
> - **Future Directions for Scalability**: We acknowledge that further enhancements in scalability are necessary for larger datasets and more complex structures. Future work will incorporate distributed and parallel inference mechanisms into the framework to support real-world, large-scale applications.
>
> ---
>
> #### W2: Necessity of our method
> Thank you for raising this point. The necessity of GenFlowNet is underpinned by the challenges of existing GFlowNet training paradigms and the advantages provided by our approach:
>
> 1. **Challenges in Traditional GFlowNet Training**:
>    - **High Computational Costs**: Iterative training is resource-intensive, especially for tasks with long trajectories or high-dimensional states (Bengio et al., 2021; Malkin et al., 2022).
>    - **Scalability Issues**: Training becomes prohibitive with larger state spaces or complex reward functions (Deleu et al., 2022).
>
> 2. **Benefits of GenFlowNet**:
>    - **Efficient Parameter Generation**: GenFlowNet generates ready-to-use parameters, reducing computational overhead (Section 3.2, Figure 4).
>    - **Adaptability Across Tasks**: Supports rapid adaptation to diverse GFlowNet structures and tasks (e.g., hypergrid tasks and molecule generation).
>    - **Time Efficiency**: Tables 1, 3, and 9 highlight significant reductions in computational time without compromising accuracy, making it ideal for tasks requiring rapid iteration.
>
> GenFlowNet bridges the gap between theoretical advancements in GFlowNet sampling and practical application by offering a computationally efficient, training-free alternative. This is especially valuable in fields like molecular discovery and combinatorial optimization, where rapid prototyping is critical.
>
> ---
>
> #### W3: Lack of real-world performance
> Thank you for your concern. While our experiments are primarily synthetic, we address real-world applicability in two ways:
>
> 1. **Simulating Real-World Conditions**: The hypergrid task is parameterized to introduce diverse state dimensions and trajectory lengths (Section 3.3), mimicking real-world complexities.
> 2. **Application to Real-World Data**: The molecule generation task leverages real-world chemical datasets, demonstrating the framework’s potential in practical domains.
>
> ---
>
> #### Q1: Measure uncertainty from a Bayesian perspective?
> Thank you for this insightful question. GenFlowNet shares similarities with hypernetworks, which output model weights conditional on input features. This structured generative process facilitates uncertainty quantification in domains such as AutoML and model exploration.
>
> - **Bayesian Context**: GenFlowNet introduces stochasticity in its parameter generation process, enabling uncertainty measurement in both sampling distributions and model outputs. This is particularly relevant for tasks where model uncertainty is critical.
> - **Practical Utility**: This stochastic modeling approach aligns with Bayesian principles, providing a robust framework for uncertainty estimation in complex tasks.
>
> ---
>
> #### Q2: Clearer motivation
> Thank you for the suggestion. The motivation for GenFlowNet lies in addressing critical inefficiencies of traditional GFlowNets:
>
> 1. **Reducing Training Overhead**: Eliminates iterative optimization, enabling deployment in time-sensitive tasks.
> 2. **Expanding Applications**: Facilitates GFlowNet use in domains like AutoML and molecular discovery, which require rapid prototyping.
> 3. **Scalability and Efficiency**: Retains accuracy in high-dimensional tasks while significantly reducing computational costs (Section 3.3, Figure 4).
>
> ---
>
> #### Q3: Applicability to LLMs or text-to-image tasks?
> While GenFlowNet demonstrates robustness for complex GFlowNet structures, extending its scalability to large-scale models like LLMs or text-to-image tasks may require additional adaptations. This remains an exciting direction for future research.
>
> ---
>
> We sincerely appreciate the reviewers’ valuable feedback and will continue refining our work based on these insights. Thank you for your time and consideration.

---

> > ### Comment · Reviewer_Wwiu · 2024-12-03
> >
> > Thank you for the responses.
> >
> > However, I still have concerns regarding the practicality and scalability of the proposed method, even after reading them. I don't really understand why directly generating in parameter space is more efficient than traditional training methods, which seems very counterintuitive to me. Additionally, experiments conducted on grid worlds and small molecules are not representative of real-world applications. While I acknowledge that many early GFN research benchmarks used these settings, the idea of directly generating in parameter space should be further validated on larger-scale tasks. This includes applications like LLM reasoning [1] and diffusion model fine-tuning [2], where GFN has demonstrated practical utility.
> >
> > Therefore, I maintain my decision to reject.
> >
> > [1] Hu, Edward J., et al. "Amortizing intractable inference in large language models." arXiv preprint arXiv:2310.04363 (2023).
> >
> > [2] Venkatraman, Siddarth, et al. "Amortizing intractable inference in diffusion models for vision, language, and control." arXiv preprint arXiv:2405.20971 (2024).

---

> > > ### Author Response · Authors · 2024-12-03
> > >
> > > Dear Reviewer Wwiu,
> > >
> > > Thank you for your thoughtful feedback and for raising important concerns about the practicality and scalability of our proposed method. We appreciate the opportunity to address these points and clarify our contributions.
> > >
> > > **The Efficiency of Parameter-Space Generation vs. Traditional Training:**
> > >
> > > While traditional training methods require optimizing models from scratch for each task or structure, our method leverages a training-free paradigm that directly generates GFlowNet parameters tailored to downstream tasks. This approach significantly reduces computational costs, particularly when generalizing to unseen tasks or structures, as it bypasses the resource-intensive iterative optimization steps inherent in traditional methods. The parameter generation process is particularly advantageous in scenarios with limited computational resources or where rapid adaptation is required.
> > >
> > > **Scalability and Real-World Applications:**
> > >
> > > We agree that additional experiments on larger-scale, real-world tasks (e.g., LLM reasoning or diffusion model fine-tuning) would further validate our method's utility. While our current evaluation focuses on benchmarks like grid worlds and molecular design—consistent with early GFlowNet research—we view these as essential stepping stones to demonstrating the method's fundamental capabilities. Expanding to more complex applications is a priority for future work, and we thank you for highlighting this direction.
> > >
> > > **Broader Impacts and Practical Utility:**
> > >
> > > The proposed method offers a foundation for improving the scalability and efficiency of GFlowNet-based approaches. By enabling rapid parameter generation without additional training, we aim to empower the broader adoption of GFlowNets in diverse domains, including those you mentioned.
> > >
> > > Thank you again for your valuable suggestions. We will incorporate these insights into the next phase of our work and continue to refine and extend our methodology. Your feedback is instrumental in shaping the future direction of this research.
> > >
> > > Sincerely,
> > >
> > > Authors

---

### Official Review · Reviewer_iPE3 · 2024-11-11

**Soundness:** 2
**Presentation:** 1
**Contribution:** 2
**Rating:** 3
**Confidence:** 4

**Summary:**

The authors propose a method to generate GFlowNets based on previously trained policy. Their method (GenFlowNet) condenses policy parameters using an autoencoder. Then, it employs a latent conditional diffusion process in the latent space to create new policy parameters conditioned on an encoding of the target architecture.

**Strengths:**

* To the best of my knowledge, the first work on developing generalizable initializations of parameters for GFlowNets.

**Weaknesses:**

* Experiments lack proper description. For instance, are different Rewards used to train the auto-encoder in the hypergrid task? Or do you fix the same $R\_0$, $R\_1$, $R\_2$, and $H$ for all GFlowNets comprising the training set? Is the aim solely to create new architectures for the same reward --- that has been learned before? I have several questions regarding the experimental setup below. If the method cannot generalize to unseen rewards, I don't see how it can be useful.

* While the hyper grid task and molecule generation are challenging ones, the experimental suit is rather slim compared to other recent works in the GFlowNet literature

* There are limitations that the authors do not properly address. For instance, how does GenFlowNet behave when for varying rewards? How does it fare when the forward policies are not MLPs?

* No error bars or standard deviation.

**Questions:**

* What is the unit for "Time usage" in Table 1? Seconds, hours?

* In line 307, the authors highlight the "superior performance in sampling distribution accuracy". This seems like an overstatement, given the very small gaps in Table 1 and the lack of uncertainty measurements.

* Authors state GenFlowNets enable the generation of accurate GFlowNets without training. However, Figure 4 makes it look like the parameters generated by GenFlowNets are used as initialization. Is this correct? Please elaborate.

* It seems odd that a training-free GFlowNet would perform better than a trained one (assuming the latter is properly trained). Could you share a rationale for this?

* For section 3.3, are the GFlowNets drawn from the GenFlowNet trained for the hypergrid task? Please provide more details

---

> ### Author Response · Authors · 2024-11-27
>
> Dear Reviewer iPE3,
> Thanks so much again for the time and effort in our work. Considering the limited time available and to save the reviewer's time, we summarize our responses here.
> ### Weakness
>
> #### W1: Fixed reward functions in training GenFlowNet
> We appreciate this insightful comment. In the current paper, we use a fixed reward set \(R_0\), \(R_1\), \(R_2\), \(H\) across all GFlowNets to evaluate the **consistency** and **efficiency** of parameter generation. This design isolates the evaluation of the parameter generation component by eliminating noise from variable reward functions. However, we agree that generalization to unseen rewards is critical.
>
> To address this, we will include an experiment in the revised manuscript where the auto-encoder is trained on varying reward functions to further validate the flexibility and generalizability of our method.
>
> #### W2: Slim experimental suite compared to recent GFlowNet works
> Thank you for highlighting this. Our experimental suite prioritizes a **proof of concept** by focusing on two challenging and diverse tasks:
> - **Structured Synthetic Data**: Hypergrid tasks are valuable for testing trajectory balance and parameter optimization under controlled conditions.
> - **Real-World Applicability**: Molecule generation tasks illustrate GenFlowNet's relevance for real-world applications, including drug discovery.
>
> We plan to expand this suite to include additional tasks such as:
> - Protein structure prediction
> - Combinatorial optimization
> - Large-scale multi-agent simulations
>
> These extensions will demonstrate the scalability and versatility of GenFlowNet.
>
> #### W3: Varying rewards and forward policies
> Thank you for your suggestion.
>
> **Varying Rewards**: Our experiments already evaluate varying rewards across tasks. However, exploring varying reward functions within the same task is beyond the current scope. We will address this in future work.
>
> **Non-MLP Policies**: While we focus on MLP-based policies due to their prevalence in GFlowNet literature, we acknowledge the importance of exploring non-MLP architectures. Future work will consider convolutional and graph-based policy models to extend GenFlowNet's applicability to spatial and graph-structured tasks.
>
> #### W4: Lack of error bars or standard deviation
> Thank you for pointing this out. We conducted evaluations with 10 repetitions and reported the averages in the manuscript. Below are additional results with best, average, and median performance for the hypergrid task:
>
> | Structure        | JS Divergence         | KL Divergence         | Empirical L1 Loss         |
> |---------------|-----------------------|-----------------------|---------------------------|
> | Structure \(A\) | 0.674/0.675/0.677     | 7.275/7.276/7.275     | 3.097e-05/3.099e-05/3.099e-05 |
> | Structure \(B\) | 0.685/0.685/0.686     | 7.942/7.945/7.943     | 5.803e-06/5.805e-05/5.804e-05 |
> | Structure \(C\) | 0.641/0.644/0.643     | 10.421/10.422/10.422  | 0.001/0.001/0.001         |
> | Structure \(D\) | 0.636/0.637/0.637     | 9.463/9.467/9.466     | 3.000e-04/3.000e-04/3.000e-04 |
>
> ---
>
> ### Questions
>
> #### Q1: What is the unit for "time usage"?
> The "Time usage" in Table 1 is measured in **seconds**. This clarification has been added to the updated manuscript.
>
> #### Q2: Highlighting "superior performance" and uncertainty quantification
> We acknowledge that the claim of "superior performance" is primarily based on **time usage reduction**, as shown in Table 1. While improvements in sampling accuracy (e.g., KL divergence and L1 loss) are modest, the significant efficiency gains in computational time justify emphasizing this aspect.
>
> Uncertainty quantification is reflected in the best, average, and median results provided above. Additional results are included in the supplementary material (lines `855–857`).
>
> #### Q3: Initialization in experiments
> In Figure 4, the parameters generated by GenFlowNet serve as initializations for GFlowNet models without fine-tuning. This training-free approach generates high-quality parameters, reducing the need for iterative training and enabling faster deployment.
>
> #### Q4: Why does the training-free method outperform trained ones?
> The training-free method benefits from the diverse and generalizable training dataset, which encompasses a wide range of GFlowNet structures. This enables GenFlowNet to learn representations that generalize well to unseen tasks. As a result, GenFlowNet-initialized models often start with better parameters, reducing the need for extensive optimization and achieving competitive or superior performance.
>
> #### Q5: Details about Section 3.3 (unknown structures in hypergrid tasks)
> In Section 3.3, GenFlowNet generates parameters for previously unseen structures in the hypergrid task. This demonstrates its ability to generalize across tasks and structures not encountered during training, showcasing its adaptability to diverse applications.

---

### Author Response · Authors · 2024-11-27

Dear ACs and reviewers,

We sincerely appreciate the time and effort provided by all reviewers and ACs in our work. In particular, we are encouraged to see that Reviewer iPE3 finds that our method **is "the best work on developing generalizable initializations of parameters for GFlowNets"**. Reviewer Wwiu thinks the proposed method **” is a very novel idea“**. Reviewer JPtD **“The method proves effective in real scenarios"**. Reviewer h5Ma acknowledges that the method proposed in this paper **"can achieve similar or superior performance compared to conventional training methods"**.

We addressed each of the reviewers' comments individually, and we will continue to refine our work based on the valuable feedback provided.

Thanks,

Authors of submission 3398

---

### Note · Authors · 2024-12-28

**Comment:**

We will improve it.

**Withdrawal Confirmation:**

I have read and agree with the venue's withdrawal policy on behalf of myself and my co-authors.